# Insect Distribution in a Vacant Multi-Level Office Building

**DOI:** 10.3390/insects14070578

**Published:** 2023-06-25

**Authors:** Peter Brimblecombe, Laure Jeannottat, Pascal Querner

**Affiliations:** 1Department of Marine Environment and Engineering, National Sun Yat-Sen University, Kaohsiung 80424, Taiwan; p.brimblecombe@uea.ac.uk; 2School of Environmental Sciences, University of East Anglia, Norwich NR4 7TJ, UK; 3Swiss National Library, Hallwylstrasse 15, 3003 Bern, Switzerland; laure.jeannottat@nb.admin.ch; 4Natural History Museum Vienna, Burgring 7, 1010 Vienna, Austria; 5Institute of Zoology, Department of Integrative Biology and Biodiversity Research, University of Natural Resources and Life Sciences Vienna (BOKU), Gregor-Mendel-Straße 33, 1180 Vienna, Austria

**Keywords:** *Tineola bisselliella*, IPM, pheromone traps, Switzerland, spatial distribution, *Zygentoma*, *Attagenus smirnovi*

## Abstract

**Simple Summary:**

Insects are a problem in domestic and commercial environments where they damage stored products, fabric, and wood. In a vacant building, during its preparation for renovation, a clothes moth (*Tineola bisselliella*) population grew particularly large on the floors close to ground level. A campaign to trap and identify the insects present suggested that the warm and stable climate allowed the insects to thrive. The basement levels, where sunlight was absent, saw large numbers of silverfish, and the population was dominated by the hitherto little-known ghost silverfish (*Ctenolepisma calvum*). The larger rooms tended to have more insects. The study suggests that in unoccupied buildings, even where food is limited, insect populations can thrive and expand, so it is important that the interior is occasionally checked for invasive pests.

**Abstract:**

The webbing clothes moth (*Tineola bisselliella*) causes extensive and costly damage to fabrics, furnishings, and museum objects. It is best known from its presence in homes, museums, and historic properties, while infestations in office buildings are not as well understood. Offices typically have more frequent cleaning, fewer quiet habitats, less food availability, and fewer breeding environments for moths, which may explain the lower abundance. Nevertheless, they can be introduced with materials or by employees whose homes have a moth infestation. This study examines the distribution of different insect pests determined from pheromone traps set out in an unoccupied multi-floor office building in Switzerland. *Tineola bisselliella* dominated the insect catch but was mostly found in the aisles on the lower floors. The larger rooms tended to have a greater insect catch. Carpet beetles (*Attagenus smirnovi*) and silverfish (*Zygentoma*) were also found, although in smaller numbers, and often preferred the basement floors. The ghost silverfish (*Ctenolepisma calvum*) dominated the *Zygentoma*, even though it has been rare until recently in Switzerland. The study suggests the need for Integrated Pest Management within office buildings. In addition, in unoccupied buildings under renovation, with no obvious sources of food, insect pests still need monitoring.

## 1. Introduction

Insect pests have long been associated with our society. Synanthropic organisms often play an important role in the ecosystem and are key in decomposing organic waste; however, many of their effects are negative, causing the destruction of crops, transmitting diseases, inducing allergic reactions in humans, and promoting damage to organic materials. An association with human habitation has resulted in a long interest in their control in the urban environment, with broad economic, social, and political implications [1,2].

In homes and residential buildings, ants (Family: *Formicidae*) and flies (Order: *Diptera*) can be a nuisance, as they are attracted to food, garbage, especially in the kitchen, and are a special concern for restaurants. Cockroaches (Order: *Blattodea*) thrive in warm, humid environments and can be found in kitchens, bathrooms, and basements. Bed bugs (Genus: *Cimex*) and mosquitoes (Family: *Culicidae*) can be a problem in homes, feeding on the blood; their bites causing itching and rash. In shops and commercial locations, stored product pests, such as beetles (Order: *Coleoptera*), weevils (Superfamily: *Curculionoidea*), and moths (Order: *Lepidoptera*), can infest dry goods, such as flour, rice, and grain. They can cause significant damage to food supplies and can be difficult to detect until an infestation is well established. In larger buildings, a range of other insects are also found, such as stink bugs (Family: *Pentatomidae*) [3]. In hotels and residences, bed bugs can be a problem, especially in the tropics [4] where they can create crises [5,6], with huge implications for hotel management. They readily disperse in high-rise buildings with the movement of residents and furnishings [7]. Insects can also be a problem for emergency rooms and hospitals [8], where flies, ants, and spiders are also to be found [9,10].

Moths have long been found in human habitations, so concern about damage to textiles goes back to biblical times (Job 4:19). The webbing clothes moth (Genus: *Tineola*) causes extensive and costly damage to fabrics and furnishings [11,12,13]. It is found throughout temperate regions with moderate humidity, in undisturbed indoor spaces. It is known as a commune pest in homes, museums, and historic properties, while infestations in commercial buildings are less discussed and poorly understood. Office buildings typically have more frequent cleaning, fewer quiet habitats, fewer food sources, and fewer breeding environments for moths, which may explain their lower abundance. The pest status of the clothes moth derives from its unusual ability to digest keratin, so the larvae may damage bird feathers, animal fur, hair and other keratinaceous materials, and woolen textiles, such as carpets, blankets, rugs, clothing, tapestries (sometimes silk, linen, cotton and synthetic fibers can also be infested; however, the moth larvae need keratin or animal wool to develop). Materials are more likely to appeal to moths when stained with food, body oil, blood, sweat, or urine; this is particularly important for the infestation of synthetic materials. Damage often becomes apparent as holes in the material and disfigurement by webbing and frass.

The clothes moth became less common in Europe through the 1970s with the increasing use of artificial fibers that are more resistant to attack. A move to mixed fibers and less aggressive use of pesticides has enabled their continued infestation. This has led to an increasing threat, with the moth being one of the most common and widely dispersed museum pests that has become an increasing problem [12,14,15]. Climate change has further been seen as a driver of changing insect populations [16,17,18,19]. They are common in cities and can infest new buildings by flying in from the surrounding area, but mostly they are transported inside with infested materials [20]. Moths can also feed on dead animals (e.g., mice) or organic materials within dust [21,22]. In the past, it was assumed that they also originated from bird nests; however, this is more likely true for the case of the bearing clothes moth (*Tinea pellionella* Linnaeus, 1758) [12,15]. However, insect pests can also be introduced by employees whose homes are infested. Silverfish (Order: *Zygentoma*) and carpet beetles (Family: *Dermestidae*) are also common pests in Europe and are a particular worry because these destroy organic materials, such as paper and furnishings in a wide variety of indoor settings [23].

This study examines the distribution of different insect pests from traps set out in a multi-floor office building in Switzerland that has four basement levels. The building had a troublesome insect infestation that provided an additional opportunity for investigation prior to renovation. This study gave a particular focus to the large number of *Tineola* sp. caught, along with additional attention to the smaller numbers of *Zygentoma* and *Dermestidae*. The study explored the potential ways of their introduction and possible environmental issues relevant to their spatial distribution. The work may provide clues to the origin of infestations and suggests solutions (treatment) for Integrated Pest Management within office buildings.

## 2. Materials and Methods

The building (Figure 1a) investigated in this study is located in a commercial district of a large city in Switzerland. The structure consists of four basement levels, a ground floor, and five upper floors, the topmost being an attic. It was mainly used as office space, with the basement used for computers and other IT technology. In 2020, the building was emptied for renovation (ground floor, 1st floor, and −1 floor basement), which is set for completion in the spring of 2023, followed by reoccupation in the autumn of 2024. Heating the building uses oil, and the cooling and ventilation systems run all the time, but at a low rate while it was virtually empty, without occupants working in the offices. The insect infestations were discovered in 2021 and investigated by monitoring, followed by a campaign aimed at eradication.

A total of 212 pheromone traps for webbing clothes moths were distributed in the building on the 6 April 2022: 125 traps were placed in rooms (both in room spaces and under the floor) and a further 87 traps in the corridors. All traps were placed on the ground to also collect other crawling insects. The catch record is derived from 4-month exposures. The traps typically use the female-produced pheromone, (E)-2-octadecenal and (2E,13Z)-2,13-octadecadienal, although the commercial traps may contain other attractants [24]. We used FINICON pheromone traps for the webbing clothes moths in our monitoring exercise.

The insects identified and discussed in this paper are: *Tineola bisselliella* Hummel, 1823 (*Tineidae*; webbing clothes moth), *Attagenus smirnovi* Zhantiev, 1973 (Dermestidae; brown carpet beetle), and silverfish or *Zygentoma*: *Lepisma saccharinum* Linnaeus, 1758 (*Lepismatidae*; common silverfish), *Ctenolepisma longicaudatum* Escherich, 1905 (*Lepismatidae*; long-tailed silverfish) [25,26], and the newly introduced, *Ctenolepisma calvum* Ritter, 1910 (*Lepismatidae*; ghost silverfish) [27]. *Ctenolepisma calvum*, originally from Ceylon (now Sri Lanka), was first found in Europe (Hungary) in 2003, though is now rapidly expanding into central Europe [25]. The adult insects trapped in the study were identified to species level. Some moth flies from the family *Psychodidae*, which includes drain flies and sewer gnats, along with book lice or psocids (Class: *Insecta*; unranked: *Paraneoptera*) were found in the traps. Additionally, there were a small number of other insects, such as ants (Family: *Formicidae*), and spiders (Order: *Araneae*), but these are not discussed within this paper.

The indoor climate was measured with a Hobo data logger from April to June of 2021 within the upper levels, from Floors 1 to 5, designated with the codes: OG1 (goods receipt area), OG2, OG3, OG4 and OG5. In the basement, the levels −1–−4: UG1, UG2, UG3, UG4 were also monitored. Additionally, outdoor climate data was extracted as 30-year climate norms (MeteoSwiss) and daily data from the Weather Underground (https://www.wunderground.com/history; both accessed on 1 March 2023).

An insect catch requires thoughtful statistical treatment as it is necessarily integer and frequently zero. We have often used both the mean and the median ($\tilde{x}$) to describe the central tendency and the lower and upper quartiles (*Q*_1_, *Q*_3_) to represent dispersion. The catch has been displayed as box-and-whisker plots, with the box showing *Q*_1_ and *Q*_3_, the median denoted by a central line and the mean a cross in the box. Whiskers show the range, except for outliers (any number that lies over 1.5 times the interquartile range beyond *Q*_1_ and *Q*_3_). Nonparametric statistical tests were used to explore the relationships between data sets (Kendall rank correlation, statistic *τ*) and the difference between data sets (Mann–Whitney test, statistic *U*_A_). The Kruskal–Wallis one-way analysis of variance was used as a non-parametric method for testing whether the daily climate observations from different floors shared the same distribution.

## 3. Results and Discussion

### 3.1. Overall Catch

More than 3500 insects were trapped in the building across the sampling period. As seen in Figure 2, the catch was dominated by *T. bisselliella* (3626). There were smaller numbers of *Attagenus smirnovi* (45) and their larvae (177), along with *Zygentoma* or the silverfish (78), the family *Psychodidae* or moth flies (25), and book lice (Psocids) (67). The high catch of *T. bisselliella* partially comes as a result of the use of pheromone traps, while for other species, it was possible that they were attracted to the dead moths, or alternatively just blundered onto the traps. The insects were so abundant that only five of the 212 traps captured no insects. The catch rate (insects/traps) was almost 19; however, the content of the traps varied widely, so the mean is not especially representative. The low relative abundance of *Psychodidae* and book lice (Psocids) should not be taken to mean that they are generally rare in urban areas. *Psychodidae* can be found in drains and sewage systems. Book lice (Psocids) are frequently found in homes and museums where they feed on detritus, but they prefer environments where the relative humidity is above 50%.

Although most traps revealed the presence of *T. bisselliella*, other insects were typically absent, which made statistical comparisons between species difficult as there were so many zero catches. Nevertheless, in the rooms, there was a weak relationship between *T. bisselliella* and *A. smirnovi* larvae (Kendall *τ* = 0.14, *p*_2_~0.04); however, such a relationship could not be found in the corridors, although fewer that 20 non-zero catches were found in these locations, making comparison difficult. The *Zygentoma* showed a weak negative relationship with *T. bisselliella* (Kendall *τ* = −0.17, *p*_2_~0.05).

Overall, 78 *Zygentoma* were caught in the building: 23 *L. saccharinum* (common silverfish), 18 *C. longicaudatum* (long-tailed silverfish), and 37 *C. calvum* (ghost silverfish). The high proportion of *C. calvum* in the catch (Figure 2b) reflects a change in indoor species that can also be seen elsewhere. This insect was rarely recorded in the past, but it is becoming more frequent in Austria, Germany, and Switzerland [27]. Figure 2c shows the proportions of *Zygentoma* found in Viennese museums [28]. *Ctenolepisma lineatum* Fabricius, 1775 (*Lepismatidae*; the four-lined silverfish) is not found in buildings in Switzerland and is still rare in Viennese museums; however, it is increasingly found across Europe [29].

### 3.2. Aisles and Rooms

As *T. bisselliella* was so abundant, it is possible to analyze the distribution in some detail and separate the traps into the moths trapped in the aisles and those in the rooms with raised floors. There were 785 adult *T. bisselliela* in traps in the rooms with suspended floors and 2841 in the aisles. The catch rate (insects/traps) was low in the rooms with 6.28 ± 7.7 ($\tilde{x}$ = 4, *Q*_1_ = 2 and *Q*_3_ = 7), while in the corridors, it was 32.7 ± 20.2 ($\tilde{x}$ = 31, *Q*_1_ = 21 and *Q*_3_ = 46). The Mann–Whitney test suggests that the catch rates in the corridors were significantly higher (*U*_A_ −9597.5; *p*_2_ < 0.0001). In addition, the distribution of insects varies between the rooms and the corridors, as shown in Figure 3. These forms might be expected if the catch was randomly distributed across the traps (as a Poisson distribution [21]).

The distributions for the other species were more difficult to establish as many traps had a zero catch and, in some cases, the species identification was difficult. The catch rates for Zygentoma were 0.15 in the rooms and 0.90 in the corridors. On the face of it, this suggests that the catch in the corridors was greater, though a Mann–Whitney test failed to show a significant difference (*p*_2_~0.29), as so many values were zero. The family *Pyroglyphidae* had catch rates of 0.064 and 0.68 in the rooms and corridors, but again, the dominance of a zero catch means it is difficult to give the result any significance. Although 25 book lice were trapped, these were all caught in the corridors. By contrast, the adult *Attagenus smirnovi* were caught at almost equal catch rates in the rooms and corridors, i.e., 0.18 and 0.26, while the larvae were caught at higher rates in the rooms than in the corridors, i.e., 1.36 and 0.08.

### 3.3. Vertical Distribution

The distribution of *T. bisselliella* changes with the floor of the building. It is especially notable in the insects trapped within the corridors (Figure 4a), where the largest numbers are found on the ground floor (Floor 0). This declines slightly with height but more dramatically below the ground floor.

In line with the observations made in the earlier paragraph, the catches in the traps were smaller in the rooms than in the corridors for Floors 5 down to Floor −1 (there is no data from rooms deeper into the basement). The Mann–Whitney test suggests that all these differences were significant at *p*_2_ < 0.0001, with the exception of the floors above (*p*_2_ < 0.02) and below (*p*_2_ < 0.06) the ground floor. The median catch in the rooms of these two floors (Figure 4b) was the highest found on any floor, but even here the catches are significantly higher in the corridors.

The catch of *Attagenus smirnovi* was lower so a similar analysis is not possible; the catch rate (summed across both adults and larvae) from the floors and corridors is shown in Figure 4c. The catch was dominated by larvae in the rooms from Floor 5 to Floor −1 where they were numerous, but few were caught in the corridors below this. The numbers seem to vary floor-to-floor but there appears to be no obvious trend. With such a small catch of adults or larvae on the individual floors, statistical uncertainty remains.

The catch of the *Zygentoma* from the floors and corridors is shown in Figure 4d and tends to be greater on the lower floors, especially on the basement floors, which have a stable environment, are quiet, and lack bright sunshine (Figure 5). The 67 examples of the book lice (Psocids) were all caught in the basement corridors, which seems to hint at parallel preferences.

### 3.4. Room Size

Implicit in some equations for insect occurrence is the idea that the number of insects caught in a room is proportional to the area [30]. This notion seems to be based on the sense that a large room might provide more food or habitats, although insects often seem to be found at the edges of a room, so the perimeter might be a better measure for the number present. Figure 6 shows the catch in various rooms as a function of the room area digitized from plans. The relationship seems very poor; although overall, the Kendall rank correlation coefficient is *τ* = 0.17 with *p*_2_ < 0.01, suggesting a significant level of correlation. However, as the rooms sampled on Floor −1 basement were larger than the typical higher floors (filled purple squares, Figure 5) and catches in the basement were comparatively high (Figure 4b), it is possible that this caused the relationship to be stronger than in reality. However, when the correlation was re-determined, omitting the basement catches, it gave a Kendall rank correlation coefficient of *τ* = 0.12 and *p*_2_~0.07; weaker, but still of modest significance.

The high scatter in Figure 6 suggests that although there is a relationship it is hard to see it as a good determinant of insect catch. Nevertheless, rooms where the catch was greater than 15 tend to be large. The data was split into large and small rooms as defined by the point shown in the inset to Figure 6 (a few rooms were omitted because of the uncertain catch, and some rooms had two traps). The median catch in the 80 smaller rooms (<20 m^2^) was three (*Q*_1_ = 2, *Q*_3_ = 5) compared with the 39 large rooms with the median of six (*Q*_1_ = 2.5, *Q*_3_ = 16.5), a difference that is significant using the Mann–Whitney test (statistic *U*_A_ = 2133, *p*_2_~0.012). The data is insufficiently detailed to distinguish whether the catch is more closely related to the room area or room perimeter.

### 3.5. Food Sources

The most abundant pests, the webbing clothes moth (*T. bisselliella*), carpet beetles (*Attagenus smirnovi*) and the *Zygentoma* are all omnivores but prefer animal-based materials, often as fabrics. Although *T. bisselliella* is usually introduced with infested objects brought into the building, they can also fly through open windows in cities. They often feed on wool textiles but can also live off dust, dead animals, or live in bird nests. In the office building studied here, it is uncertain what a large number of moths feed on. However, the food source must be sufficient and of good quality to support a population above 8000 individuals (we caught only ~4000 male moths). *Attagenus smirnovi* were also quite common in the building, where they seem to feed mainly on dead insects (Figure 7a). The former use of the building as an office space has probably resulted in large accumulations of dust and dead insects in gaps, shafts, and underneath the floorboards (Figure 7b), which provide a potential source of food.

### 3.6. Indoor Climate

The mean temperatures and relative humidity from the late spring to early summer of 2021 are shown in Figure 8. Unfortunately, the climate records from inside the building covered just over two months; however, the temperatures on the upper floors of the building were higher than those outdoors (Figure 8a), and across this period, there was an increase in temperature on the upper floors in line with the approach of the warmer summer weather and increased insolation. There was a large diurnal range to the temperatures, especially in the attic (OG5). The strong day-to-day variation had a range of more that 8 °C, and in June, the daytime temperatures sometimes exceeded 30 °C. The other floors had less extreme temperatures but were typically above 20 °C, which is sufficient for moths to fly in and to disperse through the building. Deep in the basement, the conditions were cooler and almost constant, fluctuating by just some tenths of a degree (Figure 8b). In general, the temperature in mid-May (10–20) revealed an increase from the basement floors to the attic (inset to Figure 8b). The April, May, and June temperatures for the city, in 2021, were 7.3 ± 3.9, 10.8 ± 2.3 and 18.6 ± 2.4 °C, which can be compared to the 30-year normals 1991–2020 of 9.0, 13.2 and 16.9 °C, respectively, suggesting that April and May were a little cool, but June was warmer than normal.

The relative humidity on all floors showed an increase from April to June (Figure 8c), although the rooms remained drier than the external conditions; this is not surprising given that the interior was warmer than the outside. Although the records all look rather similar, the Kruskal–Wallis test suggests that in mid-May (10–20), the relative humidity for the floors on both the upper and basement levels were not drawn from the same source (in both cases *p* < 0.0001). In particular, the two lowest floors of the basement were the most humid, although because the building is rather dry the levels are modest (Floor −3: 48.8 ± 2.3%; Floor −4: 48.8 ± 4.0%). In April, May and June of 2021, the outdoor relative humidity was 68.2 ± 9.4, 76.3 ± 11.6 and 77.6 ± 9.4%, which can be compared to the 30-year normals 1991–2020 of 70, 72 and 72%, suggesting that April was a little drier but May and June slightly damper. The absolute humidity in the city for these months of 2021 was 5.39, 7.53 and 12.38 g m^−3^, respectively. This can be compared with the values in the two lowest basement floors, which were about 8.2 g m^−3^.

Figure 8d–g shows the relationships between the spring temperature, relative humidity, and the catch rate for *T. bisselliella* and the *Zygentoma*. These hint at different responses from these insects, with the *Zygentoma* preferring the lower cooler and damper floors, while *T. bisselliella* preferred the warmer and slightly less humid floors above ground. Although trends seem satisfying, it is not clear that the temperature and RH were the key drivers, as it could well be sunlight, food availability or the level of disturbance. An example of this would be the availability of water; although the building is certainly dry compared with the external environment, there is a stream flowing in the basement that might provide water (Figure 9).

The external climate in April of 2022 (8.7 ± 2.5 °C) was, thus, more than a degree warmer than in 2021 and a little more humid (73.3 ± 9.2%), but this would hardly suggest that the building would not be amenable to insect growth. In the summer, temperatures in the attic fluctuated between day and night but even the night-time lows were ~22 °C; therefore, suitable for insect growth. The relative humidity was moderate, and although insects would prefer 50–60%, somewhat higher than the range found in the building, it would be unlikely to create difficulties for their survival. Particularly as they could seek out micro-habitats where the conditions were a little damper. There was a slight preference for the ground and first floor of the building where the moth catch was high but this was likely to have been driven by the route through which they entered the building, rather than the climate. The *Zygentoma*, by contrast, seemed somewhat more abundant in the basement and may have preferred dark conditions. Being rather shy, they may benefit from this quiet and uninterrupted environment, along with the flowing water (Figure 9). Although only a small number (67) of psocids were found, these were all from basement floors with humid conditions they prefer.

### 3.7. Summary Discussion and Intervention

Moths dominated the catch, although the pheromone traps were specifically tuned to trap them. Among the *Zygentoma*, *C. calvum* dominated, which may be a reflection of a general change seen among silverfish species in Austria, Germany, and Switzerland [27]. The trapping campaign showed that the insects were largely caught on the floors close to ground level, which could hint that this was the initial route of entry, most food was available here, or the climate conditions were the best. The largest numbers were typically found trapped in the corridors, rather than in the rooms or in the underfloor spaces. There appeared to be a high number of moths caught on traps in the larger rooms, although the relationship was not strong enough to establish whether this was a function of the room area or perimeter. The conditions in the building, although a little dry, were mostly favorable to the insects present. The *Zygentoma* seemed to prefer the basement floors in stable conditions and a lack of sunlight may account for this preference. A similar result was found for *Ptinidae* caught in Nikkō in Japan, with the largest numbers trapped in the basements [31].

The source of the infestation remained unclear but there were large quantities of dust and animal remains, accumulated over many years, in the underfloor spaces that provided a likely food source. However, the traps laid there did not catch more moths than the traps in the corridors (Figure 3). This serves as a reminder that, even in unoccupied buildings, it is possible for pests to seek out sufficient food to support a widely distributed infestation. Additionally, low occupancy can be a time for shy insects, such as the *Zygentoma*, to range more widely [32].

The high level of infestation in the empty building posed a risk for its future use as offices, reading rooms, and temporary storage spaces. Managing such infestations represents a major challenge for Integrated Pest Management, especially when the source of the infestation remains undetermined. A range of methods were considered for the eradication of the pests:(i)In areas of high activity of the three silverfish species, *L. saccharinum, C. longicaudatum* and *C. calvum*, an application of Advion bait gel (active ingredient: 0.6% indoxacarb) was suggested [25,26]. However, treating the whole building using this method was not an option because of its size.(ii)Large numbers of *Attagenus smirnovi* were found under some floors. It suggested that these spaces needed deep-cleaning and possibly the application of a contact insecticide, e.g., pyrethroids or desiccant dust. However, cleaning these dead spaces would be costly and time consuming. A test cleaning of one room revealed little reduction in the numbers trapped.(iii)The largest problem arose from the *T. bisselliella* caught mostly in the corridors. Treatment with sulfuryl fluoride is not an option in Switzerland, and although fumigation with a contact insecticide product was also discussed, without locating the source of the infestation, the large volumes needed might still be ineffective. In the end, parasitoid wasps were released in areas with large infestations to attack the larvae [33].

The situation will continue to be monitored for pest activity and, if needed, further wasps may be released. As the renovated building will be back in use by the end of 2024, it is important to mitigate the risks to the materials to be stored in the renovated and adapted building. It is especially important that materials taken out of the building do not transport pests to other storage facilities.

## 4. Conclusions

*Tineaola bisselliella*, the *Zygentoma,* and *Attagenus smirnovi* infested an unoccupied multi-floor building. These clothes moths, silverfish, and carpet beetles were able to thrive despite the dry conditions and a lack of obvious food, such as textiles made from animal wool or dead animals. They may have found damper microclimates and found sufficient food in the dust and detritus in underfloor or other hard to clean spaces. The observations are a reminder that undisturbed places represent a habitat for insect pests. Future work should examine the food sources for such infestations and determine why they might prefer corridors or larger rooms. It also suggests the need for simple monitoring techniques that might provide warning of infestations developing in buildings under renovation.

## Figures and Tables

**Figure 1 insects-14-00578-f001:**
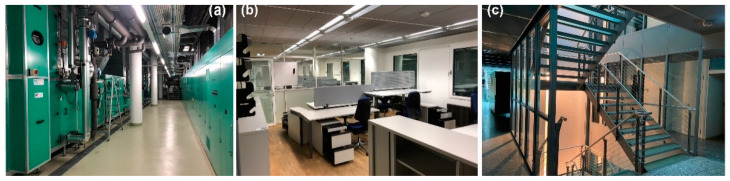
The building interior: (**a**) plant room in the basement, (**b**) office spaces and (**c**) stairwell and corridors. Photo credits: Pascal Querner.

**Figure 2 insects-14-00578-f002:**
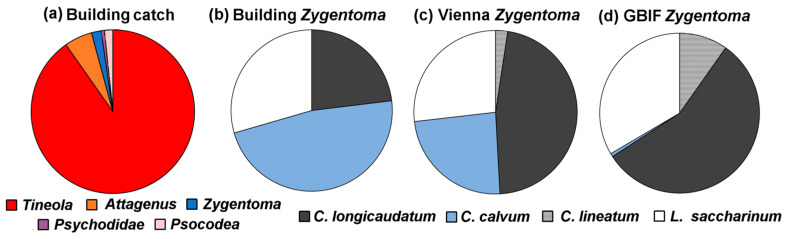
(**a**) Distribution of insect catch in the building. (**b**) Distribution of the *Zygentoma* catch in the building. (**c**) Distribution of the *Zygentoma* catch from Viennese museums [27]. (**d**) Distribution of the four *Zygentoma* species as exported in Europe from the year 2000 onwards in the Global Biodiversity Information Facility (GBIF) https://www.gbif.org/, accessed on 1 March 2023.

**Figure 3 insects-14-00578-f003:**
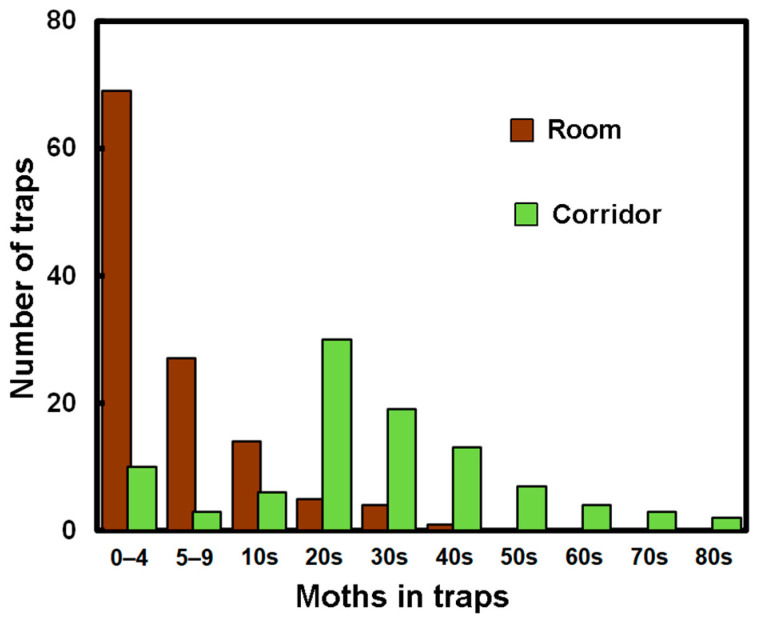
Distribution of *Tineola bisselliella* in rooms and corridors.

**Figure 4 insects-14-00578-f004:**
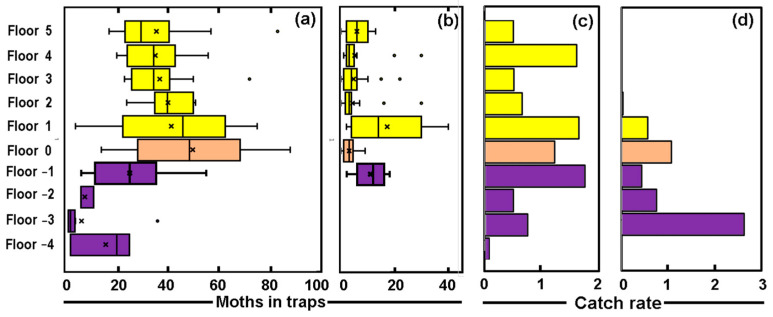
Vertical distribution of the (**a**) catch of *Tineola bisselliella* in corridors, (**b**) catch of *T. bisselliella* in rooms, (**c**) catch rate of *Attagenus smirnovi* and (**d**) catch rate of the *Zygentoma*. Note: yellow denotes upper floors; pink the ground floors; and violet the basement.

**Figure 5 insects-14-00578-f005:**
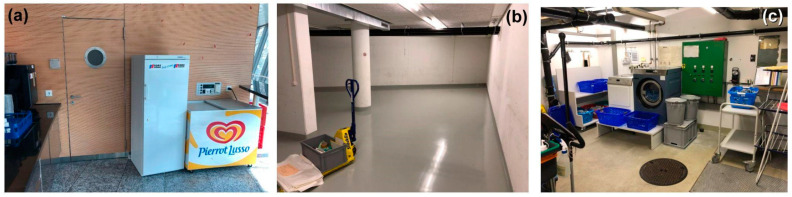
Floors with large silverfish catches: (**a**) first floor with the former canteen, (**b**) the quarantine area, and (**c**) the cleaning room in the basement. Photo credits: Pascal Querner.

**Figure 6 insects-14-00578-f006:**
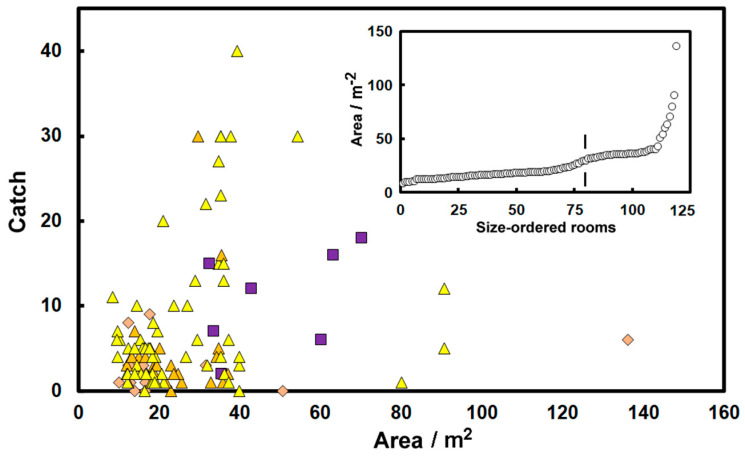
Catch of *Tineola bisselliella* as a function of room area. The inset shows the areas of individual rooms arranged in order, small to large, where the vertical line marks the point that distinguishes large rooms from small rooms. Note that triangles represent upper floors, diamonds the ground floor, and squares the basement.

**Figure 7 insects-14-00578-f007:**
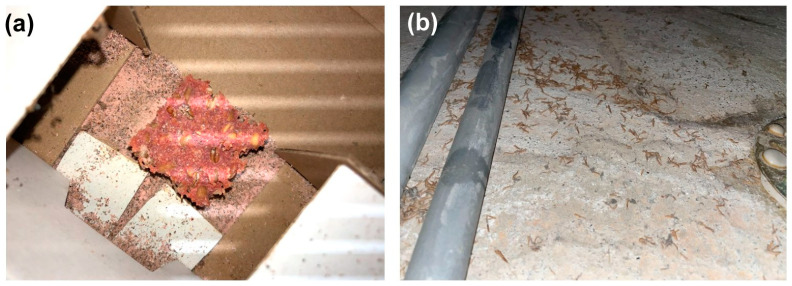
Infestation of an old poison mouse bait by brown carpet beetles (*Attagenus smirnovi*) (**a**) and a dead space underneath the floor of the office rooms where dust had accumulated (**b**). Photo credits: Pascal Querner.

**Figure 8 insects-14-00578-f008:**
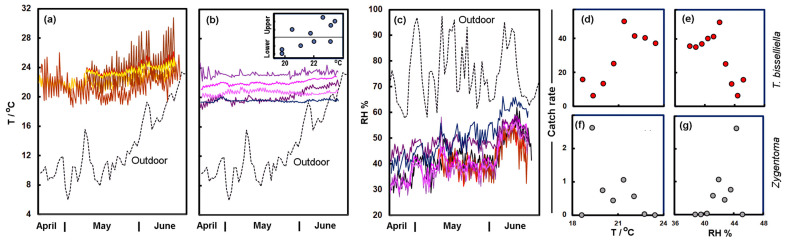
(**a**) Daily average temperature (dotted line) compared with the temperature of the upper floors. (**b**) Temperature compared with the temperature on the basement floors. The inset shows the average temperatures, 10–20 May, for the Floors −4 to 5, e.g., the basement to the attic. (**c**) Daily average RH compared with the RH of the upper and basement floors. (**d**) Catch rate of *Tineola bisselliella* as a function of the spring temperature on each floor. (**e**) *Tineola bisselliella* catch rate as a function of the spring relative humidity. (**f**) *Zygentoma* catch rate as a function of the spring temperature. (**g**) *Zygentoma* catch rate as a function of the spring relative humidity.

**Figure 9 insects-14-00578-f009:**
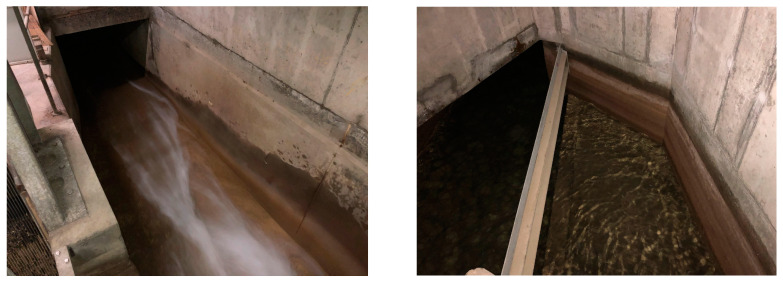
Subterranean stream flowing underneath the building’s basement. Photo credits: Pascal Querner.

## Data Availability

The insect catch can be requested as an Excel file from the authors.

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
