# Peer review of "Insect Distribution in a Vacant Multi-Level Office Building"

_insects, 2023, doi:10.3390/insects14070578_

Round 1

Reviewer 1 Report

Judging from the submissions to Insects which I have the privilege to review, the journal is increasingly becoming a publication venue for unusual, unorthodox and somehow punkish articles - which is not criticism at all, such articles deserve existence and publications, because it is the unorthodox, which triggers general progress. I therefore enjoyed reading your manuscript a lot, concluding that you did highly interesting work. 

So much being said, keep in mind that unorthodox data require the more orthodox processing and presentation. Otherwise, you risk producing junk, which will "pass" into a journal only because reviewers like the bravery. In this case, there are several serios flaws, which can be improved by careful revision, including more rigorous statistical analysis. I am a bit afraid that some of the analyses may require that you return to the building and collect additional data, as external predictors. 

My major comments are 
1. that you did not number the lines! (it makes communication quite difficult) 
2. The statistics. Your approach is not necessarily wrong, but you are tapping too little from your data. To reitarate, you had huge number of traps, with "community" samples, characterised by such parameters as floor - room or corrditor - room size - even climate. Such data beg for a multivariate ordination (principal components, canonical corresponence...) which would overcome the problems with statistical distributions, allow you nice visualisations of the results, and in the same time, allow you direct testing of such hypothesis, as floor effect, room size effect... etc. By re-visiting the building, you may also obtain semiquantitative data on food supply, which would further strengthen your conclusion.  

Many more remarks I found while reading your text: 

Introduction

general: You state, at the end, that you will investigate "different species", but devote entire paragraph to single species, Tineola bisselliella. The text needs reframing, e.g., that you investigate many species, paying particular attention to one of them. 

2nd para (and elsewhere): You should realise that it is scientific paper, and despite the fresh style of the writing, which I am enjoying a lot, stick to such rules as stating scientific names of organisms (cockroaches, bedbugs...) when you mention them for the first time. 

3rd para, Tineola bisselliella. Scientific name of a species has 4 parts, genus, species, authority and year of description. At the first mention, state full names. 

food cources = food sources

4th para: "The moth has been associated for thousands of years" - associated with whom? I assume that with humans, but your readers are not expeted to make assumptions. 

Material and methods
Why do you keep "the large city" secret? I believe that no-one will recognise (and care) where the building is. 

By further reading, I see (from figure 1) that the city was Bern. But the legend "Bern Zygentoma" does not make sense without informing reader that the city was Bern!

page 3: "the newly introduced, though rapidly expanding, Ctenolepisma calvum Ritter, 1910" - from where it was introduced? could be incorporated into the sentence.

Results 

General. I am not sure about the policy of the "Insects" journal, but my life-long experience says that Results section should be written in simple past tense, if possible. Present tense implies patterns that are general, independent of a study, which surely is not the case of your statements at page 5 (below) and 6 (top). Please, re-write. 

Food sources section
"throzugh" = through 

The whole section is more interpretative than presenting results. Did you try to quantify the food sources somehow? Even a semiquantitative description of the sources (e.g. carpentry, clothes etc. present/absent) might allow an interesting analysis. 

Plus, I dislike your "population estimation". Catch of whatever number of moths into traps cannot inform you about population size (it will depend, inter alia, on efficiency of the traps, which will likely decline with exposition time). Use the number of 8000 as a minimum estimate, not as an estimate. 

Indoor climate. Although interesting, these are again more Interpretations than results. As a minimum, I would recommend to relate the climate on the floors to catches of individual species, via a correlation analysis. But an ordination analysis would suit your results better. 

Discussion, p 15

Here, the Discussion is hardly a discussion, as required by a scientific text. It is, rather, a series of recommendation, "how to kill the nasty bugs", which is OK for a company report, but not for general readers. 

Notably, you wrote some of the general entomology interpretations already to Results section, and so, you can improve the Discussion rather simply, by switching parts of the text (and reducing the "kill the bugs" recommendations to absolute minimum. 

"an application of the Advion bait gel (active ingredient: 0.6% indoxacarb) is surgested," = is suggested

There are multiple typos, sometimes embarrassing, so a thorough check will help.

Author Response

Author's Reply to the Review Report - Reviewer 1

Judging from the submissions to Insects which I have the privilege to review, the journal is increasingly becoming a publication venue for unusual, unorthodox and somehow punkish articles - which is not criticism at all, such articles deserve existence and publications, because it is the unorthodox, which triggers general progress. I therefore enjoyed reading your manuscript a lot, concluding that you did highly interesting work. 

.So much being said, keep in mind that unorthodox data require the more orthodox processing and presentation. Otherwise, you risk producing junk, which will "pass" into a journal only because reviewers like the bravery. In this case, there are several serios flaws, which can be improved by careful revision, including more rigorous statistical analysis. I am a bit afraid that some of the analyses may require that you return to the building and collect additional data, as external predictors. 

REPONSE Sadly not possible to return to the building as it is now under renovation, and it was a one-off opportunity. Attempts at treatment have also altered the building ecology, but we would like to go back and study the renovated building t see how it has changed. Statistical Issues will be detailed later.

My major comments are 
1. that you did not number the lines! (it makes communication quite difficult) SORRY!
2. The statistics. Your approach is not necessarily wrong, but you are tapping too little from your data. To reitarate, you had huge number of traps, with "community" samples, characterised by such parameters as floor - room or corrditor - room size - even climate. Such data beg for a multivariate ordination (principal components, canonical corresponence...) which would overcome the problems with statistical distributions, allow you nice visualisations of the results, and in the same time, allow you direct testing of such hypothesis, as floor effect, room size effect... etc. By re-visiting the building, you may also obtain semiquantitative data on food supply, which would further strengthen your conclusion.  

RESPONSE. Agree, it would have been nice to have such visualisations and we initially  started to draw up contour maps of the building, but they were not very revealing. On the issue of multivariate statistics two problems occur in our data. (i) Many of the traps had no insects (apart from Tineola) and zeroes create a problem for PCA. Studies, both in environmental (species counts for example) and non-environmental fields, highlight issues due to zero-inflated data in standard PCA and there are relevant assumptions underlying  PCA that are relevant: http://alexhwilliams.info/itsneuronalblog/2016/03/27/pca/ (ii) The data is not paired, so we do not have the temperature and RH for every room in parallel. However we have tried this loosely in a recent paper where there are many monitors in a single room [Brimblecombe, P et al  (2022). Thermohygrometric Climate, Insects and Fungi in the Klosterneuburg Monastic Library. Heritage, 5(4), 4228-4244.] and will try this for four Viennese libraries in a MS in preparation with Katharina Derksen.  We have managed to correlate room area with catch (paired data available) in Fig. 6, but we were a little disappointed. This has inspired us to analyse room size and catch from the Kunsthistorisches Museum, which has an 8-year data set and a large number of rooms. This will be ready for publication later this summer.

REPONSE Actually the source of infestation is tough, so determining the source of food is also hard, and it may be one of quality rather than quantity. Although we may return to the building once it re-opens, but following renovation everything will be much changed.

Many more remarks I found while reading your text: 

Introduction

general: You state, at the end, that you will investigate "different species", but devote entire paragraph to single species, Tineola bisselliella. The text needs reframing, e.g., that you investigate many species, paying particular attention to one of them. 

GOOD POINT. This is done now at several points in the introduction and a mention of some lesser emphasis on  Zygentoma and Dermestidae

2nd para (and elsewhere): You should realise that it is scientific paper, and despite the fresh style of the writing, which I am enjoying a lot, stick to such rules as stating scientific names of organisms (cockroaches, bedbugs...) when you mention them for the first time. 

RESPONSE: Good point, but we formally introduced the species in the method section with the binomial, discover and date. None insects such as bed-bugs, ants etc which are not mentioned again, so although these now have scientific names added, it is only in a shortened form (just as the genus, family or order).

3rd para, Tineola bisselliella. Scientific name of a species has 4 parts, genus, species, authority and year of description. At the first mention, state full names. 

RESPONSE: this was done in the method section. So now this only at genus level at the arlier  points as a general mention, with the full name to species level in the method section.

THANKS! food cources NOW food sources

 4th para: "The moth has been associated for thousands of years" - associated with whom? I assume that with humans, but your readers are not expeted to make assumptions. 

RESPONSE: Totally agree but was actually deleted during revision

Material and methods
Why do you keep "the large city" secret? I believe that no-one will recognise (and care) where the building is. 

RESONSE We agree, but the owner has insisted on this!

By further reading, I see (from figure 1) that the city was Bern. But the legend "Bern Zygentoma" does not make sense without informing reader that the city was Bern!

YES YOU ARE CORRECT! However while the paper was being reviewed the owner came back and insisted “Bern” be removed from the figures.

page 3: "the newly introduced, though rapidly expanding, Ctenolepisma calvum Ritter, 1910" - from where it was introduced? could be incorporated into the sentence.

RESPONSE: Now added details “Ctenolepisma calvum, originally from Ceylon (now Sri Lanka), was the first found in Europe (Hungary) in 2003, though is now rapidly expanding in central Europe [25].”

Results 

General. I am not sure about the policy of the "Insects" journal, but my life-long experience says that Results section should be written in simple past tense, if possible. Present tense implies patterns that are general, independent of a study, which surely is not the case of your statements at page 5 (below) and 6 (top). Please, re-write. 

 AGREE ABSOLUTELY WITH THIS PRINCIPAL SO HAVE CHECKED THROUGHOUT AND CHANGED WHERE NECESSARY.

Food sources section
"throzugh" = through YES!!

The whole section is more interpretative than presenting results. Did you try to quantify the food sources somehow? Even a semiquantitative description of the sources (e.g. carpentry, clothes etc. present/absent) might allow an interesting analysis. 

RESPONSE: yes it is interpretative, so the section has been changed to Results and Discusson

Plus, I dislike your "population estimation". Catch of whatever number of moths into traps cannot inform you about population size (it will depend, inter alia, on efficiency of the traps, which will likely decline with exposition time). Use the number of 8000 as a minimum estimate, not as an estimate. 

RESPONSE: agree which is why we have used “above 8000 individuals”.  However, we had avoided the word “estimate” in the MS for this reason, though have now tried to ensure that we don’t make such an  implication in the text

Indoor climate. Although interesting, these are again more Interpretations than results. As a minimum, I would recommend to relate the climate on the floors to catches of individual species, via a correlation analysis. But an ordination analysis would suit your results better. 

 RESPONSE:  Done now, though the relationship is not that great, though visually appealing as Figs 8d-g.

Discussion, p 15

Here, the Discussion is hardly a discussion, as required by a scientific text. It is, rather, a series of recommendation, "how to kill the nasty bugs", which is OK for a company report, but not for general readers. 

Notably, you wrote some of the general entomology interpretations already to Results section, and so, you can improve the Discussion rather simply, by switching parts of the text (and reducing the "kill the bugs" recommendations to absolute minimum. 

RESPONSE. Shortened as suggested but when we don’t add such matters and the outcome of the work  referees in the past have complained about the lack of practical suggestions.

"an application of the Advion bait gel (active ingredient: 0.6% indoxacarb) is surgested," = is suggested

DONE!!!

There are multiple typos, sometimes embarrassing, so a thorough check will help.

DONE OUR BEST.

Reviewer 2 Report

This is a nice research and deserve quick publication. I read this MS carefully and is convinced by its presentation. 

Author Response

Author's Reply to the Review Report - Reviewer 2

This is a nice research and deserve quick publication. I read this MS carefully and is convinced by its presentation. 

We thank the reviewer 2 for his support of the paper!

Round 2

Reviewer 1 Report

I am quite satisfiied for the changes and reorganisations you have done, and would vote for "accept", if not for one simple sentence in the conclusion - in the sentence 

"...were able to thrive despite dry conditions and obvious a lack of food." 

Do your data somehow imply lack of food in the building? As far as I know, any non-synthetic cloths are food for Tineola bisselliella, and any cellulose-containing matter, including dust particles, are food for silverfish. So, re-think this once more, and either support this argument, or delete it. 

further, minor

page 2, last para: Tinea or Tineolla?

list of references, the new one, Brimblecombe et al. 2013: missing space in front of title.

Additional round of revision is needed.

Author Response

We thank the Reviewer 1 for the comments and changed them according to you surgestions:

See conclusion and Tineola marked in yellow

References were addapted.
